# Identification of Novel Tumor Pyroptosis-Related Antigens and Pyroptosis Subtypes for Developing mRNA Vaccines in Pancreatic Adenocarcinoma

**DOI:** 10.3390/biomedicines12040726

**Published:** 2024-03-25

**Authors:** Qiaowei Lin, Li Liang, Qing Wang, Xiao Wang, Yang You, Yefei Rong, Yuhong Zhou, Xi Guo

**Affiliations:** 1Department of General Surgery, Zhongshan Hospital, Fudan University, Shanghai 200032, China; 17211210041@fudan.edu.cn (Q.L.); rong.yefei@zs-hospital.sh.cn (Y.R.); 2Medical Oncology department of Xiamen Branch, Zhongshan Hospital, Fudan University, Xiamen 361015, China; liang.li@zs-hospital.sh.cn; 3Department of Medical Oncology, Zhongshan Hospital, Fudan University, Shanghai 200032, China; wang.qing2@zs-hospital.sh.cn (Q.W.); wangxiaoapril@foxmail.com (X.W.); you.yang@zs-hospital.sh.cn (Y.Y.)

**Keywords:** mRNA vaccine, pancreatic adenocarcinoma, pyroptosis subtype, pyroptosis landscape, hub genes

## Abstract

Background: As one of the important components of immunotherapies, mRNA vaccines have displayed promising clinical outcomes in solid tumors. Nonetheless, their efficacy remains unclear in pancreatic adenocarcinoma (PAAD). Given the interaction of pyroptosis with anticancer immunity, our study aims to identify pyroptosis-related antigens for mRNA vaccine development and discern eligible candidates for vaccination. Methods: Utilizing gene expression data from TCGA and ICGC, we integrated RNA-seq data and compared genetic alterations through cBioPortal. Differential gene expressions were integrated using GEPIA. Relationships between immune cell abundance and tumor antigens were analyzed and visualized via TIMER. WGCNA facilitated the clustering of pyroptosis-related genes, identification of hub genes, and pathway enrichment analyses. Pyroptosis landscape was depicted through graph learning-based dimensional reduction. Results: Four overexpressed and mutant pyroptosis-related genes associated with poor prognosis were identified as potential antigens for mRNA vaccines in PAAD, including *ANO6*, *PAK2*, *CHMP2B*, and *RAB5A*. These genes displayed positive associations with antigen-presenting cells. PAAD patients were stratified into three pyroptosis subtypes. Notably, the PS3 subtype, characterized by a lower mutation count and TMB, exhibited “cold” immunological traits and superior survival compared to other subtypes. The pyroptosis landscape exhibited considerable heterogeneity among individuals. Furthermore, the turquoise module emerged as an independent prognostic indicator and patients with high expressions of hub genes might not be suitable candidates for mRNA vaccination. Conclusions: In PAAD, *ANO6*, *PAK2*, *CHMP2B*, and *RAB5A* are prospective pyroptosis-related antigens for mRNA vaccine development, which holds potential benefits for patients classified as PS3 and those with diminished hub gene expressions, providing insights into personalized mRNA vaccine strategies.

## 1. Introduction

Pancreatic adenocarcinoma (PAAD), which was the fourth leading cause of cancer-associated death in the United States, resulted in 49,830 new deaths in 2022 [1]. Characterized by its high malignancy and poor prognosis, PAAD possesses almost equivalent morbidity and mortality, with a 5-year survival rate of only 11% [1]. Surgery is considered to be the only curative therapy for PAAD. However, the majority of PAAD patients present with local late-stage or distant metastasis at the time of diagnosis, and even those who underwent curative surgery have disease recurrence rates of nearly 90%, at a median of 7–9 months [2,3]. Standard chemotherapy prolongs life span only modestly [4,5]. Exceptionally, in less than 1% of patients with microsatellite instability high tumors, PAAD is well-known for its immunosuppressive tumor microenvironment, with the characteristics of a prominent myeloid cell infiltration and absent or dysfunctional adaptive T cell immunity, making it almost entirely refractory to immunotherapies such as immune checkpoint inhibitors (ICIs) [6,7]. Therefore, there is an urgent need to find out a new therapy to improve the prognosis of PAAD patients.

With a huge success in preventing the COVID-19 pandemic, messenger RNA (mRNA) vaccines are attracting widespread interests in both cancers and infectious disease fields [8,9]. After vaccination, mRNA vaccines will express tumor antigens, which will be recognized and taken up by antigen-presenting cells (APCs), facilitating APC activation and innate/adaptive immune stimulation [10]. Based on the antigen form, cancer vaccines can be divided into four types: peptide-based vaccines, nucleic acid-based vaccines, tumor/immune cell-based vaccines and viral vector-based vaccines [11]. Nucleic acid-based vaccines are thought to be a relatively more promising platform, allowing simultaneous delivery of multiple tumor antigens and encoding full-length tumor antigens, and they thus are more likely to stimulate a broader adaptive immune response [10,12]. Unlike DNA vaccines integrating into the tumor cell genome and causing insertional mutations, mRNA vaccines have potent translation efficiency and are non-integrating and therefore pose no genetic threats [10]. Although there are no Food and Drug Administration-approved mRNA vaccines for tumors nowadays, some successful attempts of cancer vaccines, either as monotherapy or in combination with ICIs, have been applied in various solid tumors [13,14]. In stage III/IV melanoma patients, an mRNA vaccine alone or combined with ICIs induced a powerful immune response, resulting in better prognosis of extended disease-free survival [15,16]. Recently, a preliminary phase 1 trial confirmed that an individualized mRNA vaccine, autogene cevumeran, in combination with atezolizumab and mFOLFIRINOX induces substantial T cell activity in surgically resected PAAD patients that correlates with delayed recurrence [17]. Despite substantial progress, several challenges considering the immunogenicity and effectiveness of mRNA vaccines remain, exerting influence on our clinical practices. Thus, it is of vital importance to identify individual tumor-specific neoantigens.

Pyroptosis, a kind of inflammatory type of non-apoptotic regulated cell death, is charactered by cell swelling, lysis, and the release of several proinflammatory factors such as IL- 1β, IL-18, and HMGB1 [18]. Along with the increase of these proinflammatory factors, many immunostimulatory and tumor suppressor genes are upregulated, whereas various immunosuppressive and tumor-promoting genes are downregulated [19]. There is a positive feedback loop between pyroptosis and immune cells, that is, tumor cells can release danger signals that recruit antitumor immune cells through pyroptosis while immune cells can induce pyroptosis in tumor cells [19,20]. An immunologically cold tumor is considered to be naturally refractory to ICIs. However, immunologically cold tumors were efficiently killed by adding pyroptosis inducers to ICIs while being ineffectively killed with pyroptosis inducers alone, emphasizing the importance of ICIs with pyroptosis inducers for the treatment of immunologically cold tumors [19]. Classically, PAAD exhibits an immunologically cold tumor microenvironment. In order to improve the efficacy of immunotherapy and thus further extend the life expectancy of PAAD patients, we explored pyroptosis-related genes as novel PAAD antigens for developing mRNA vaccines, defined 3 pyroptosis subtypes, and mapped the pyroptosis landscape of PAAD to select potentially suitable patients for vaccination with the help of data from The Cancer Genome Atlas (TCGA) and International Cancer Genome Consortium (ICGC).

## 2. Materials and Methods

### 2.1. Data Collection and Preprocessing

The PAAD data set, Genomic Data Commons (GDC) TCGA PAAD (*n* = 182) were downloaded from the University of California Santa Cruz (UCSC) Xena database (https://xena.ucsc.edu/) (accessed on 15 July 2023). The data type was selected as FPKM, and the “Primary solid tumor” (01A) was extracted and converted to the TPM format. The data of “Masked Somatic Mutation” were selected as the somatic mutation data of PAAD patients. The VarScan software (version 2) was used for data preprocessing. The R package ‘maftools’ was used for the visualization of somatic mutation data of PAAD patients. Meanwhile, the clinical data, including age, tumor node metastasis (TNM) stage, survival status, and survival time were achieved after eliminating patients lacking clinical data, leaving 222 patients with survival information and 176 patients with clinical data. The gene expression data from Homo sapiens and clinical data (including survival status and time) of another PAAD data set, PACA-CA, were downloaded from the ICGC database (https://dcc.icgc.org/) (accessed on 15 July 2023). After excluding patients without clinical and survival data, a total of 122 tumor samples were incorporated into this study.

In addition, the pyroptosis-related genes with scores greater than 0.15 were extracted from GeneCards (https://www.genecards.org/) (accessed on 15 July 2023), finally reaching 403 pyroptosis-related genes (Appendix A). The genes of immune cell death (ICD) and immune checkpoint (ICP) were obtained from the previous literature (Appendix A) [21,22]. A detailed flow chart is shown in Appendix A.

### 2.2. cBioPortal Analysis and GEPIA Analysis

The cBio Cancer Genomics Portal (cBioPortal, http://www.cbioportal.org) (accessed on 15 July 2023) [23] was used to integrate RNA-seq raw data from databases such as TCGA, compare genetic changes in PAAD, and extract microsatellite instability (MSI) and tumor mutation burden (TMB) data of TCGA-PAAD patients. The Gene Expression Profiling Interactive Analysis (GEPIA, http://gepia2.cancer-pku.cn) (accessed on 15 July 2023) [24] was used to integrate differential gene expressions, which were identified by analysis of variance (ANOVA) with |log2FC| values > 1 and *q* values < 0.01. The Kaplan–Meier curves were used to evaluate overall survival (OS) and progression-free survival (PFS), with the median as the cutoff value and log-rank tests for comparison. *p*-values < 0.05 were considered to be statistically significant.

### 2.3. Tumor Antigens and Immune Cells Infiltration

The analysis modules such as gene expressions, somatic mutations, clinical outcomes, and somatic copy number changes were analyzed and the relationships between tumor immune infiltration cell (TIIC) abundance and potential tumor antigens were visualized through Tumor Immune Estimation Resource (TIMER, https://cistrome.shinyapps.io/timer/) (accessed on 15 July 2023) [25]. The Spearman correlation analysis was used, and *p*-values < 0.05 were considered to be statistically significant.

Moreover, the gene expression data of the infiltration of immune cells and other stromal cells were evaluated by using the R package ‘MCPCounter’ [26], which provided the abundance estimates for eight immune cells, namely T cells, CD8+ T cells, natural killer (NK) cells, B lymphocytes, monocytic lineage, dendritic cells, neutrophils, and cytotoxic lymphocytes as well as two non-immune stromal cells, namely fibroblasts and endothelial cells. The Spearman correlation analysis based on the abundance estimates by R package ‘MCPCounter’ and the expressions of antigen genes was performed, and *p*-values < 0.05 were considered to be statistically significant.

### 2.4. Discovery and Validation of Pyroptosis Subtypes

The R package ‘ConsensusClusterPlus’ [27] was performed on cluster pyroptosis-related genes and used to construct a consistent matrix and identify the corresponding pyroptosis subtypes and gene modules on the basis of gene expression profiles. Partitioning was performed using the median algorithm of “1-Pearson correlation” distance metrics, with 1000 repetitions and resampling 80% of patients in the queue each time. The cluster sets ranged from 2 to 9, and the optimal partition was defined by evaluating the consensus matrixes and the consensus cumulative distribution functions. The pyroptosis subtypes were then validated in an independent PACA-CA cohort with the same settings.

### 2.5. Prognostic Evaluation of the Pyroptosis Subtypes

The log-rank test was used to evaluate the prognostic role of different pyroptosis subtypes. ANOVA was performed to determine the correlation between pyroptosis subtypes and various pyroptosis-related molecular and cellular features. The chi-square test was used to screen for the most frequent genetic mutations. Single-sample Gene Set Enrichment Analysis (ssGSEA) was performed to calculate the immune enrichment scores for each sample through the R package ‘GSVA’ [28].

### 2.6. Weighted Gene Co-Expression Network Analysis (WGCNA)

The modules of pyroptosis-related genes were screened using the R package ‘WGCNA’ [29]. The soft threshold was calculated through ‘pickSoftThreshold’ functions, with 4 calculated as the best soft threshold. Scale-free networks were established based on the soft threshold, followed by the construction of topological matrixes, hierarchical clustering, and eigengene calculations. The inter-module correlations were established in the light of the eigengene module, and then hierarchical clustering was performed. GSEA was performed with the help of the Metascape database (www.metascape.org/) (accessed on 15 July 2023) [30], including the Kyoto Encyclopedia of Genes and Genomes (KEGG) and Gene Ontology (GO) [31,32]. Adjusted *p*-values < 0.05 were considered to be statistically significant in the KEGG and GO analysis.

### 2.7. Construction of Pyroptosis Landscape of Tumor Microenvironment

A graph learning-based dimensionality reduction analysis was used, exploiting the dimensionality reduction capabilities of the Monocle package, which is an R package with a Gaussian distribution, in order to visualize the distribution of pyroptosis subtypes in individual patients. The maximum number of components was set to 2, and the discriminative dimensionality reduction method of ‘DDRtree’ was used. Finally, the pyroptosis landscape was visualized by using functional diagram cell trajectory of pyroptosis subtypes coded by colors.

### 2.8. Statistical Analysis

Statistical analyses were conducted utilizing R software (version 4.1.1). Differences between two groups were compared with the Wilcoxon rank sum test. Difference comparisons among more than two groups were evaluated using the Kruskal–Wallis test. Correlation analysis was performed using the Spearman correlation analysis. *p*-values < 0.05 were considered to be statistically significant.

## 3. Results

### 3.1. Identification of Potential Pyroptosis-Related Antigens in PAAD

To identify potential PAAD mRNA vaccines, we first screened for abnormally expressed genes and detected 9221 differential genes, of which 1532 were overexpressed genes that potentially encoded tumor-related antigens (Figure 1A). A total of 10,101 mutant genes were then screened by the analyses of the mutation fragments and counts of genes that potentially encoded tumor-specific antigens in individual samples (Figure 1B,C). Mutation analysis confirmed *TP53* as the most common mutation gene in terms of both mutation fragments and counts (Figure 1D,E). In addition, regardless of mutation quantity and frequency, we observed obvious mutations such as *TP53*, *CDKN2A*, *TTN*, *CKDN2A-DT*, *LRP1B*, etc., which might be of vital importance to the tumorigenesis and development of PAAD. In the combination of the results of mutation genes and overexpressed genes, we found 726 frequently mutated and overexpressed tumor-related genes that might serve as potential tumor antigens.

Subsequently, we focused on the possibility of pyroptosis-related genes as mRNA antigens. We finally identified 54 pyroptosis-related genes from the 726 potential tumor antigens above, of which 5 genes were closely associated with the OS while 4 genes were associated with PFS in PAAD after performing Kaplan–Meier curves (Figure 2A). In addition, we found that patients with high expressions of *ANO6*, *PAK2*, *CHMP2B*, and *RAB5A* genes had worse OS and PFS than those with low expressions (Figure 2B–I). Considering the interaction between tumor antigens and APCs, we further evaluated the correlation of these potential antigens with APCs by combining MCPCounter with the TIMER database. MCPCounter showed that *ANO6*, *PAK2*, *CHMP2B*, and *RAB5A* genes were positively correlated with various immune cells and stromal cells (Figure 3A), which were further confirmed by the TIMER database, especially various APCs (B cells, macrophages, and dendritic cells) (Figure 3B–E). The above results indicated that the identified tumor antigens might be directly recognized, processed, and presented by APCs to T cells, thus triggering immune responses. Meanwhile, these genes might induce pyroptosis in tumor cells, with the ability of further tumor killing.

### 3.2. Identification of Potential Pyroptosis Subtypes in PAAD

Recently, pyroptosis has been proven to play a crucial role in killing tumors and is closely associated with tumor immunity. Pyroptosis typing can help to reflect the pyroptosis status of tumors and their microenvironment, thus aiding with the identification of PAAD patients suitable for vaccination. Therefore, the expression profiles of 403 pyroptosis-related genes in TCGA-PAAD were analyzed to construct consensus clustering. On the basis of cumulative distribution functions and functional delta area, we selected k as 3, where pyroptosis-related genes were stably clustered (Figure 4A,B). The principal component analysis (PCA) showed that three pyroptosis subtypes, which were defined as PS1, PS2, and PS3, had a clear degree of differentiation (Figure 4C). Prognosis analysis manifested that PS3 had better OS than PS1 and PS2 in the TCGA-PAAD cohort (Figure 4D). Consistent with the TCGA-PAAD cohort, the PACA-CA cohort could also be divided into the pyroptosis subtypes of PS1, PS2, and PS3 (Figure 4E). The three pyroptosis subtypes in the PACA-CA cohort also showed significant prognostic differences, with PS3 having a better prognosis, while PS1 and PS2 had a poorer prognosis (Figure 4F). However, their differences were relatively small compared to those of the TCGA-PAAD cohort (Figure 4F). As for the different TNM stages, three pyroptosis subtypes showed irregular distribution, with the overall pattern of higher proportions of early-stage patients in the PS3 subtype while there were higher proportions of median- and late-stage patients in the PS1 and PS2 subtypes (Figure 4G,H). Overall, pyroptosis typing could be used to predict the prognosis of PAAD patients, which had been validated by different cohorts.

### 3.3. The Relationship between Pyroptosis Subtypes and Mutation Status

For the fact that higher TMB and somatic cell mutation rates correlated with stronger anti-tumor immunity, we calculated the TMB, MSI, and mutation counts for each patient using the mutation data of the TCGA-PAAD cohort and compared all the pyroptosis subtypes. Mutation counts and TMB were the lowest in the PS3 subtype while being relatively higher in the PS1 and PS2 subtypes (Figure 5A,B). However, there were no significant differences in MSI between the three pyroptosis subtypes, with the PS3 subtype being slightly higher than the PS1 and PS2 subtypes (Figure 5C). In addition, 30 genes, including *TP53*, *KRAS*, *CDKN2A*, etc., also exhibited different mutation statuses among different subtypes (Figure 5D). These results showed that TMB and mutation count might serve as potential indicators for using mRNA vaccines, and different pyroptosis subtypes had different mutation characteristics.

### 3.4. The Relationship between Pyroptosis Subtypes and Immunomodulator

Previous studies have shown that ICP regulators, such as *PD-L1* and *TIM3*, and ICD regulators, such as *CALR* and *HMGB1*, play critical roles in regulating host anti-tumor immunity and thus affect the efficacy of mRNA vaccines. While pyroptosis was closely associated with immune regulation, we evaluated the different expressions of ICP and ICD regulators in the three pyroptosis subtypes. We detected 25 ICD genes and 46 ICP genes both in the TCGA-PAAD cohort and PACA-CA cohort. We found that the expressions of ICD genes almost had the same trend in different subtypes, such as *ANXA1*, *CALR*, *CXCL10*, *EIF2AK2*, *HMGB1*, *MET*, *PANX1*, etc. (Figure 6A,B). In the TCGA-PAAD cohort, we observed more different expressions of ICP genes, which might be related to the larger sample numbers in this cohort. The expression trends of ICP genes were similar in the two cohorts, such as *CD274*, *CD276*, *CD44*, etc., with the highest expressions in the PS1 subtype, followed by PS2, and the lowest expressions in the PS3 subtype (Figure 6C,D). In conclusion, pyroptosis typing could reflect the expression levels of ICD and ICP regulators and could be used as a biomarker for mRNA vaccines.

### 3.5. The Molecular and Cellular Features of Pyroptosis Subtypes

Due to the fact that the response to mRNA vaccines depends on the immune status of the tumor, we scored the previously reported 28 signature genes in the TCGA-PAAD cohort and PACA-CA cohort through ssGSEA to further describe the immune cell compositions in the three pyroptosis subtypes. The immune cell compositions varied greatly in the three pyroptosis subtypes (Figure 7A,B). For instance, in the TCGA-PAAD cohort, the scores of activated CD4 T cells in PS1 were obviously higher than those in PS2 and PS3, whereas several immune cell scores, such as activated CD8 T cells, monocytes, etc., were higher in PS2 than in PS1 and PS3, which were consistent with the immune cell infiltration trends observed in the PACA-CA cohort (Appendix A). Kaplan–Meier curves showed that only central memory CD8 T cells, NK cells, and type 2 helper T (Th2) cells of 22 immune cells had prognostic differences, with higher cell scores indicating worse prognosis (Figure 7C–E). Significant differences were found in the gene expressions of central memory CD8 T cells among three pyroptosis subtypes in the TCGA-PAAD cohort while no differences were observed in the PACA-CA cohort (Figure 7F). Gene expressions of NK cells showed significant differences among three pyroptosis subtypes in both cohorts, with the highest expressions in PS2, followed by PS1 and PS3 (Figure 7G). Gene expressions of Th2 cells also exhibited significant differences in both cohorts, with the highest expressions in PS1, followed by PS2 and PS3 (Figure 7H). Based on the ESTIAMTE algorithm calculation of the immune scores, we observed that PS3 subtype had lower overall immune and stromal cell infiltrations and higher tumor purity compared with PS1 and PS2 subtypes in the TCGA-PAAD cohort (Appendix A). However, the differences were not obvious among three subtypes in the PACA-CA cohort, with higher immune scores only found in the PS2 subtype than the PS3 subtype (Appendix A). In summary, the PS2 and PS1 subtypes might represent immunologically “hot” tumors while PS3 subtype was an immunologically “cold” tumor. These results reflected the immune status of different PAAD pyroptosis subtypes, helping to identify appropriate PAAD patients for mRNA vaccination, which might induce immune infiltration in immunologically “cold” PS3 subtype PAAD patients.

### 3.6. Pyroptosis Landscape of PAAD

The pyroptosis landscape was constructed using the pyroptosis-related gene expression profiles of individual PAAD patients (Figure 8A). The x-axis (PCA1) was associated with various immune cells, of which the activated B cells, eosinophils, macrophages, mast cells, T follicular helper cells, etc. had the highest positive correlation, while the y-axis (PCA2) had a positive correlation with almost all immune cells (Figure 8B). Meanwhile, in a single subtype, we also observed some intra-cluster heterogeneity. All of the samples were further divided into six states (1, 2, 3, 5, 6, 7) according to the sample trajectory (Figure 8C). On the basis of their positions, we chose states that were located at the endpoints for further analysis, namely state 1, 3, 5, 6, and 7, and their proportions in the three pyroptosis subtypes were displayed in Figure 8D. Kaplan–Meier curves showed significant differences in OS between 5 states, of which state 6 had the best prognosis, whereas the survival curves of state 1, 3, 5, and 7 were relatively concentrated (Figure 8E). Subsequently, we compared the differences of 28 immune cells in these states and found that immune cell scores varied greatly between different states (Figure 8F). On the whole, the pyroptosis landscape based on pyroptosis subtypes did not have the ability to accurately identify the pyroptosis status and predict the prognosis of every PAAD patient. More samples may be needed for further confirmation and typing verification.

### 3.7. Identification of Pyroptosis Gene Co-Expression Modules and Hub Genes in PAAD

The identification of hub pyroptosis-related genes can help oncologists determine whether a patient is suitable for mRNA vaccine. To identify these hub genes, we constructed a WGCNA of pyroptosis-related genes, with a soft threshold of 4 for the scale-free network (Figure 9A). The gene matrix was then converted to an adjacency matrix. Each gene module was set to have at least 20 genes. After calculating the eigengenes of each module and integrating similar modules, we finally obtained 11 modules, of which the grey module represented unassigned genes (Figure 9B,C). Different modules showed different module scores, with the PS3 subtype module having the lowest scores overall (Figure 9D). Only the eigengene in the turquoise module was found to be an independent prognostic factor after using multiple Cox regression and the elimination of collinearity factors (Figure 9E). Further Kaplan–Meier curves showed that patients with lower turquoise module scores had better OS than those with higher scores (Figure 9F). Moreover, genes in the turquoise module, including *CBL*, *LYZ*, *ISG20L2*, *PPARA*, *IGF1R*, *EIF2AK2*, *BMP8A*, *CMTM3*, *LRP1*, etc., were significantly enriched in immune-related functions and pathways such as the Cytokine Signaling in Immune System, Positive Regulation of Protein Phosphorylation, and Cytokine–Cytokine Receptor Interaction pathways (Appendix A). Therefore, the hub genes can serve as biological markers for predicting the prognosis of PAAD patients and finding suitable patients for mRNA vaccines.

## 4. Discussion

PAAD, which is the sixth leading cause of cancer-associated death in China, remains a dramatic clinical challenge for pancreatic doctors, with a persistently increasing tendency towards morbidity and mortality [33]. PAAD is known to be naturally refractory to ICIs, which may be partially explained by the low rate of mutations for generating neoantigens [34]. However, some researches showed that PAAD indeed had more neoantigens than we had previously expected [35,36]. As immunogenic neoantigens have the capacity to activate T cells to induce immune responses, treatments with neoantigens may stimulate neoantigen-specific T cells and improve the prognosis of PAAD patients [17]. Pyroptosis is a kind of non-apoptotic regulated cell death, which is believed to have broad crosstalk with anticancer immunity [18]. For instance, a variety of chemotherapy drugs such as doxorubicin and epirubicin could induce pyroptosis in breast cancer by promoting the expression of nuclear PD-L1 and GSDMC and facilitating the activation of caspase-8 [37]; thus, the induction of non-apoptotic regulated cell death, such as pyroptosis, can be considered as a new anticancer therapy since cancers are born with resistance to apoptosis. Therefore, in order to improve the prognosis of PAAD patients, we established potential targets for mRNA vaccines and searched for PAAD patients that may benefit from those vaccines.

Through the overlap of overexpressed, frequently mutant, and pyroptosis-related genes, we obtained 54 genes, of which 4 genes were closely associated with the OS and PFS in PAAD, namely *ANO6*, *PAK2*, *CHMP2B*, and *RAB5A*. These four genes had exhibited a favorable correlation with APCs. This notable association positions them as compelling candidates for mRNA vaccines, with the potential to augment immune cell infiltration and instigate heightened tumor cell pyroptosis. Some of these candidates have been confirmed to play critical roles in the development and progression of PAAD. For example, *RAB5A* was overexpressed in PAAD and could promote aggressive biological behavior through regulation of the Wnt/β-catenin signaling pathway [38]. Moreover, *RAB5A* could drive PAAD subtype-dependent modulation of endosome trafficking [39] and promote the formation of filopodia via the activation of cdc42 and β1-integrin [40]. However, the functions of the other genes, *ANO6*, *PAK2*, and *CHMP2B*, have not undergone investigation as of present. This signifies the need for further experimental endeavors aimed at delineating the precise roles played by these three genes in the context of PAAD. Such endeavors are crucial in establishing a robust theoretical foundation to underpin the development of mRNA vaccines.

Given the pervasive nature of tumor heterogeneity, the efficacy of mRNA vaccines is anticipated to be limited to a subset of patients. To discern the most suitable candidates for this therapeutic approach, a stratification strategy was employed among PAAD patients. This categorization involved segregating patients into three distinct pyroptosis subtypes (PS1, PS2, PS3), which was contingent upon the distinctive expression profiles exhibited by pyroptosis-related genes. Kaplan–Meier curves manifested that PAAD patients classified as PS3 had better OS than PS1 and PS2 both in the TCGA-PAAD and PACA-CA cohorts, which might be related to the outcome of low mutation count and TMB in the PS3 subtype. These results indicated that pyroptosis typing could be used for the prediction of the prognosis of PAAD patients. Apart from the prediction of prognosis, pyroptosis typing is also indicative of the therapeutic effect of mRNA vaccines. With the least expression of various ICD and ICP genes and the least overall infiltrations of immune and stromal cells, the PS3 subtype typically exhibited an immunologically “cold” tumor and was less likely to respond to ICIs. Currently, it is a great challenge for an immunologically “cold” tumor to benefit from ICIs. However, mRNA vaccines, as a promising method of cancer immunotherapy, may have the capacity of converting cultivated or barren land on steep slopes into grassland and forests, namely the reinvigoration of the immune system and stimulation of more immune infiltration to execute the antitumor effects. In some preclinical studies, mRNA vaccines were used in immunologically “cold” tumors, and promising outcomes were achieved in some solid tumors, such as pancreatic cancer, renal clear cell carcinoma [41], prostate cancer [42], etc. On the contrary, the PS1 and PS2 subtypes might represent immunologically “hot” tumors with more immune cell infiltrations and higher immune scores. Meanwhile, several important ICP genes, such as *CD274*, *CTLA4*, *CD276*, etc. were highly expressed in the PS1 and PS2 subtypes, indicating an immunosuppressive tumor microenvironment in these patients, and they were more likely to be benefit from ICIs. The combination of mRNA vaccines with ICIs might be a theoretically successful strategy for these patients. Higher immune cell scores of central memory CD8 T cells, NK cells, and Th2 cells correlating with worse prognosis further confirmed the immunosuppressive tumor microenvironment in PAAD. Collectively, the pyroptosis typing of PAAD in this study not only has shown prognostic relevance but also could serve as an indicator to choose suitable patients for mRNA vaccines or combination therapies. However, the complex pyroptosis landscape of PAAD failed to accurately identify the pyroptosis status and predict the prognosis of every PAAD patient, which was an obstacle to the success of personalized mRNA vaccines. More samples were needed for further confirmation and typing verification. Furthermore, *CBL*, *LYZ*, *ISG20L2*, *PPARA*, *IGF1R*, *EIF2AK2*, *BMP8A*, *CMTM3*, *LRP1*, etc. were determined as hub genes clustered in the turquoise module, and their upregulation correlated with worse prognosis, suggesting that PAAD patients with high expression of these hub genes were not suitable for this kind of mRNA vaccine. Notably, hub pyroptosis-related genes were significantly enriched in immune-associated functions and pathways, including Cytokine Signaling in Immune System, Cytokine–Cytokine Receptor Interaction pathways, etc., further emphasizing the broad interaction between pyroptosis and immune systems.

Our findings have provided a novel treatment option for PAAD, but there are some limitations. For example, although we use data from two databases, in vitro validation of mRNA vaccines is needed to strengthen our study. Moreover, more samples are needed to confirm and verify our typing system.

## 5. Conclusions

In summary, we identified four pyroptosis-related genes as potential PAAD antigens for mRNA vaccine development, namely *ANO6*, *PAK2*, *CHMP2B*, and *RAB5A*. The mRNA vaccine may be beneficial for patients of PS3 and for patients with low expressions of hub genes. Our research provides a theoretical foundation for the development of mRNA vaccines against PAAD and screens for the optimal candidates for vaccination.

## Figures and Tables

**Figure 1 biomedicines-12-00726-f001:**
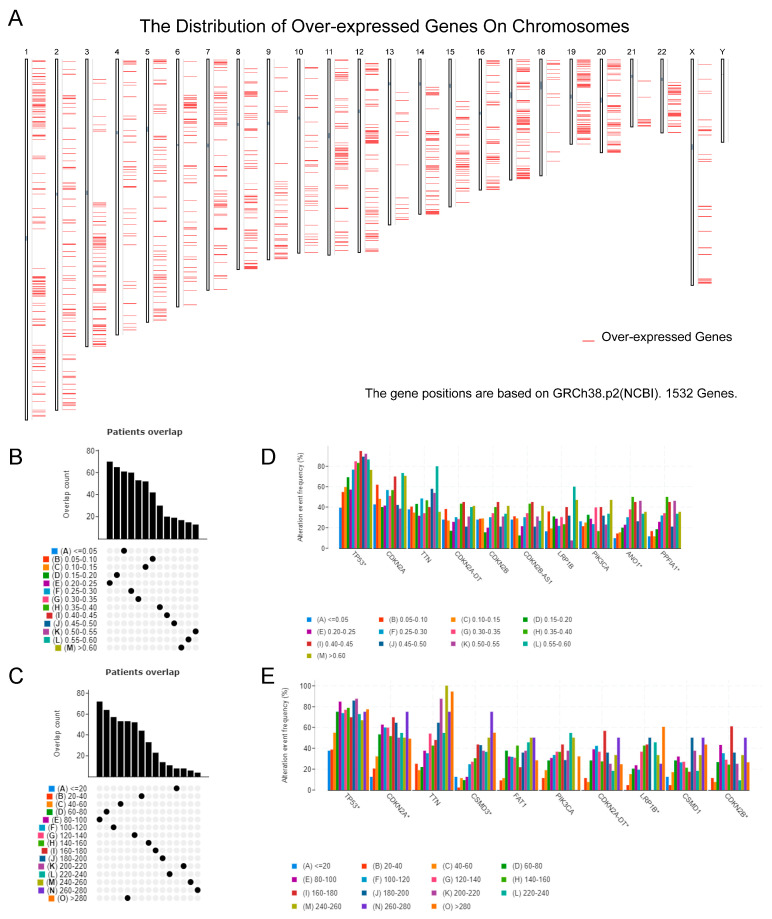
Identification of potential tumor antigens in PAAD. (**A**) The chromosomal distribution of upregulated genes in PAAD to identify potential tumor-associated antigens. (**B**–**E**) Identification of potential tumor-specific antigens in PAAD. Samples overlapping in (**B**) altered genome fractions and (**C**) mutation counts. Genes with the highest frequency in (**D**) altered genome fractions and (**E**) mutation counts.

**Figure 2 biomedicines-12-00726-f002:**
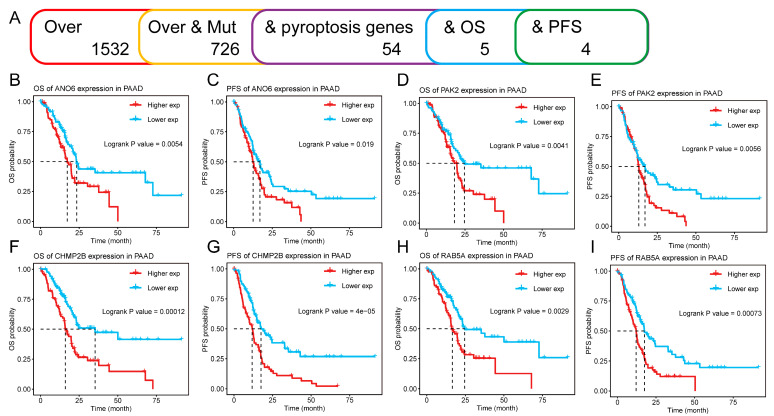
Identification of tumor antigens associated with the prognosis of PAAD patients. (**A**) Screening process of potential tumor antigens. (**B**–**I**) Kaplan–Meier curves of OS and PFS in PAAD patients classified by the gene expression levels of *ANO6* (**B**,**C**), *PAK2* (**D**,**E**), *CHMP2B* (**F**,**G**), and *RAB5A* (**H**,**I**).

**Figure 3 biomedicines-12-00726-f003:**
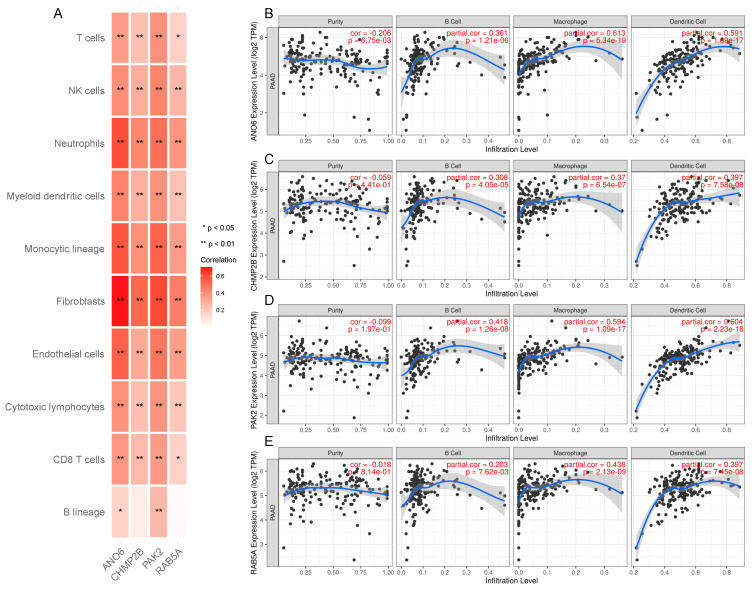
Identification of tumor antigens correlated to APCs. (**A**) The correlation analysis of immune and stromal cells with *ANO6*, *PAK2*, *CHMP2B*, *RAB5A* calculated by MCPcounter. (**B**) Correlation of the expression levels of *ANO6* (**B**), *PAK2* (**C**), *CHMP2B* (**D**), *RAB5A* (**E**) with the infiltration of B cells, macrophages, dendritic cells, and the purity of the tumor.

**Figure 4 biomedicines-12-00726-f004:**
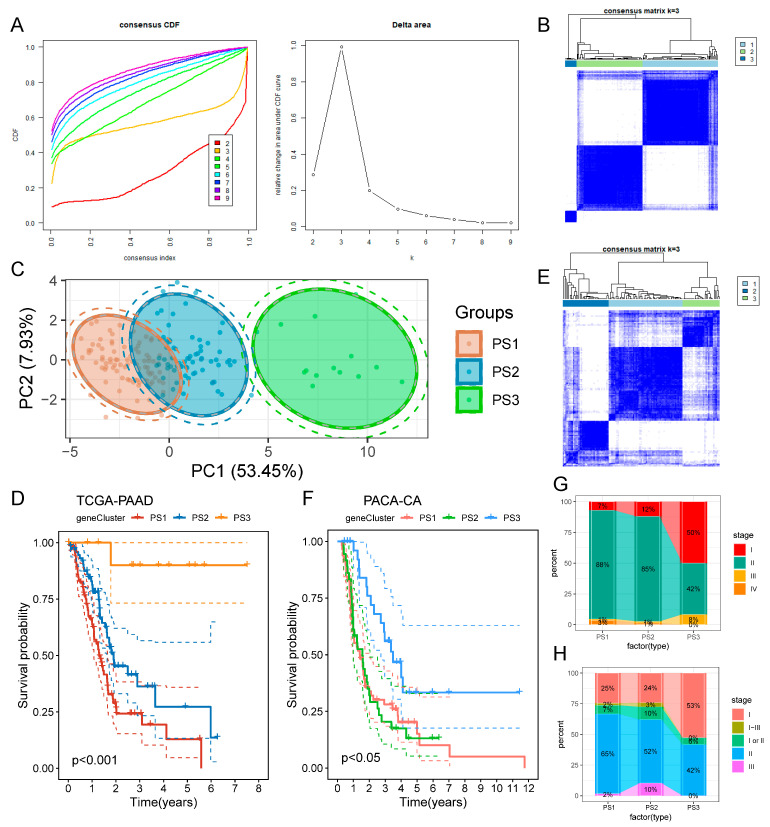
Identification of potential pyroptosis subtypes in PAAD. (**A**) Cumulative distribution functions and functional delta area of pyroptosis-related genes in the TCGA-PAAD cohort. (**B**) Clustering heatmap of TCGA-PAAD samples (k = 3). (**C**) Principal component analysis of pyroptosis subtypes in the TCGA-PAAD cohort. (**D**) Kaplan–Meier curves of OS in different pyroptosis subtypes in the TCGA-PAAD cohort. (**E**) Clustering heatmap of PACA-CA samples (k = 3). (**F**) Kaplan–Meier curves of OS in different pyroptosis subtypes in the PACA-CA cohort. (**G**) Proportional distribution of PAAD stages in different pyroptosis subtypes in the TCGA-PAAD cohort. (**H**) Proportional distribution of PAAD stages in different pyroptosis subtypes in the PACA-CA cohort.

**Figure 5 biomedicines-12-00726-f005:**
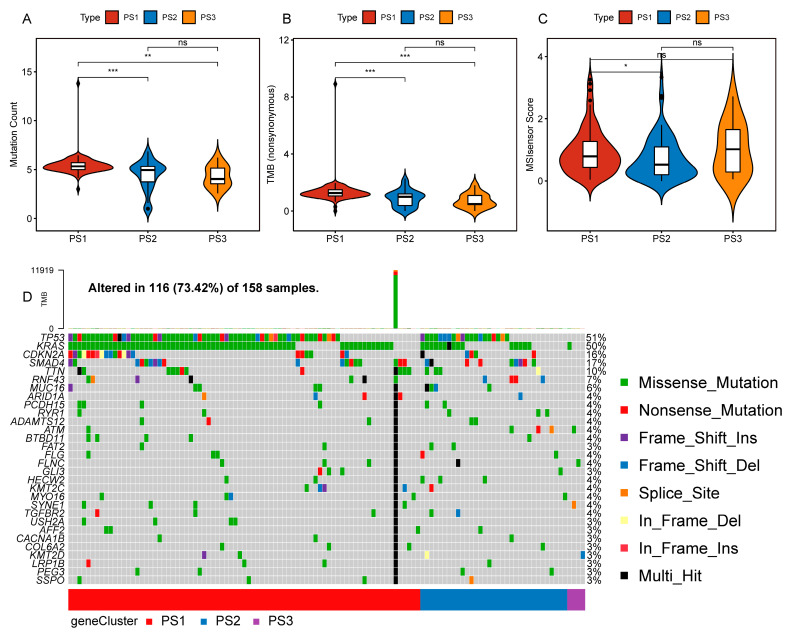
Relationship between pyroptosis subtypes and mutation status. (**A**–**C**). Comparison of mutation count (**A**), TMB (**B**), and MSI (**C**) among different pyroptosis subtypes in PAAD (ns means nonsense). (**D**). Waterfall chart of characteristic mutation genes in three pyroptosis subtypes (PS1, PS2, and PS3). * *p* < 0.05, ** *p* < 0.01, *** *p* < 0.001.

**Figure 6 biomedicines-12-00726-f006:**
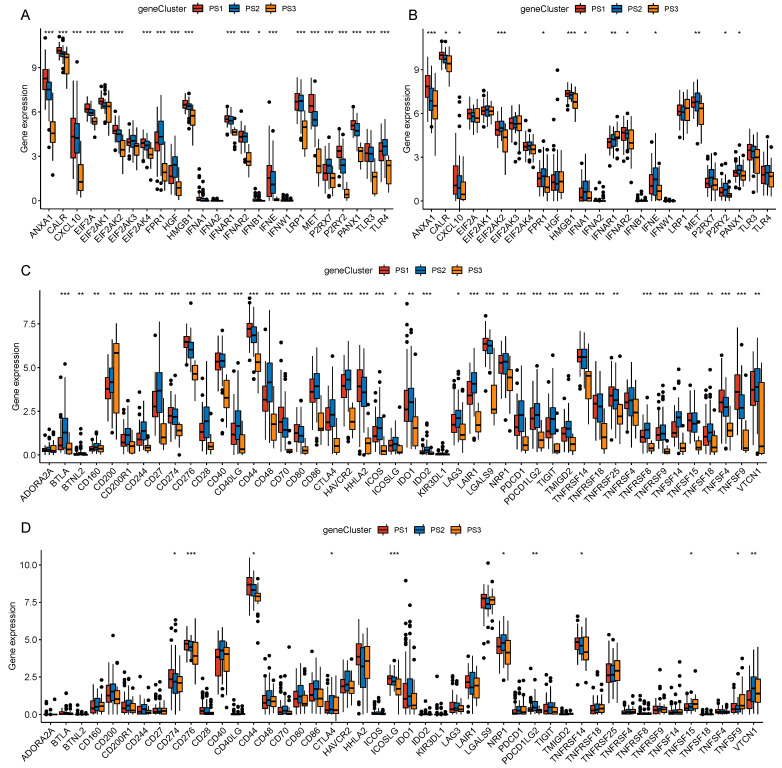
Relationship between pyroptosis subtypes and immunomodulator. (**A**,**B**) Comparison of the expressions of ICD regulators in the TCGA-PAAD cohort (**A**) and PACA-CA cohort (**B**) in three pyroptosis subtypes. (**C**,**D**) Comparison of the expressions of ICP regulators in the TCGA-PAAD cohort (**C**) and PACA-CA cohort (**D**) in three pyroptosis subtypes. * *p* < 0.05, ** *p* < 0.01, *** *p* < 0.001.

**Figure 7 biomedicines-12-00726-f007:**
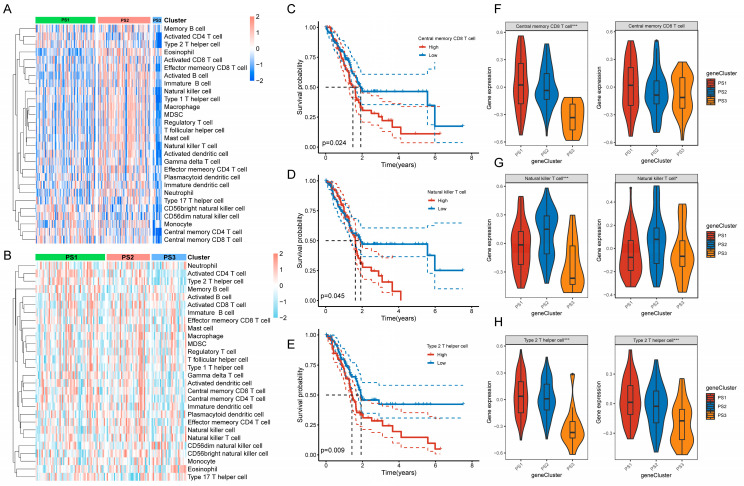
Molecular and cellular characteristics of pyroptosis subtypes. (**A**,**B**) Heatmap of enrichment scores of 28 immune cells in the TCGA-PAAD (**A**) cohort and PACA-CA (**B**) cohort among different pyroptosis subtypes in PAAD. (**C**–**E**) Kaplan–Meier curves of OS in the TCGA-PAAD cohort classified by the immune scores of central memory CD8 T cells (**C**), NK cells (**D**), and Th2 cells (**E**). (**F**–**H**) Differences in enrichment scores of central memory CD8 T cells (**F**), NK cells (**G**), and Th2 cells (**H**) among three pyroptosis subtypes in the TCGA-PAAD cohort and PACA-CA cohort. * *p* < 0.05, *** *p* < 0.001.

**Figure 8 biomedicines-12-00726-f008:**
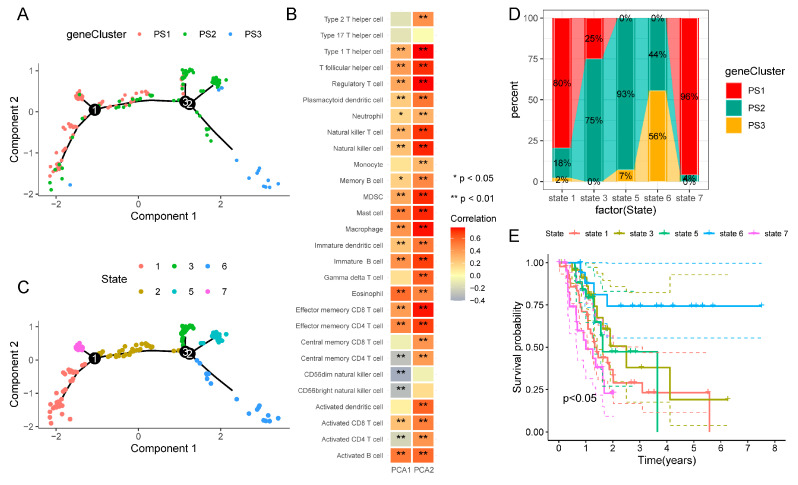
Pyroptosis landscape of PAAD. (**A**). Pyroptosis landscape of PAAD. Each dot in the landscape represents a patient and three colors correlated to three pyroptosis subtypes, representing the overall characteristics of the pyroptosis-related microenvironment. (**B**) Correlation between PCA1/2 and immune cells. (**C**) Reclassification of PAAD patients according to their locations. Different colors represent different states. (**D**) Proportional distribution of three pyroptosis subtypes in different states. (**E**) Kaplan–Meier curves of OS in different states. (**F**) Comparison of gene expressions of different immune cells in five different states. * *p* < 0.05, ** *p* < 0.01, *** *p* < 0.001.

**Figure 9 biomedicines-12-00726-f009:**
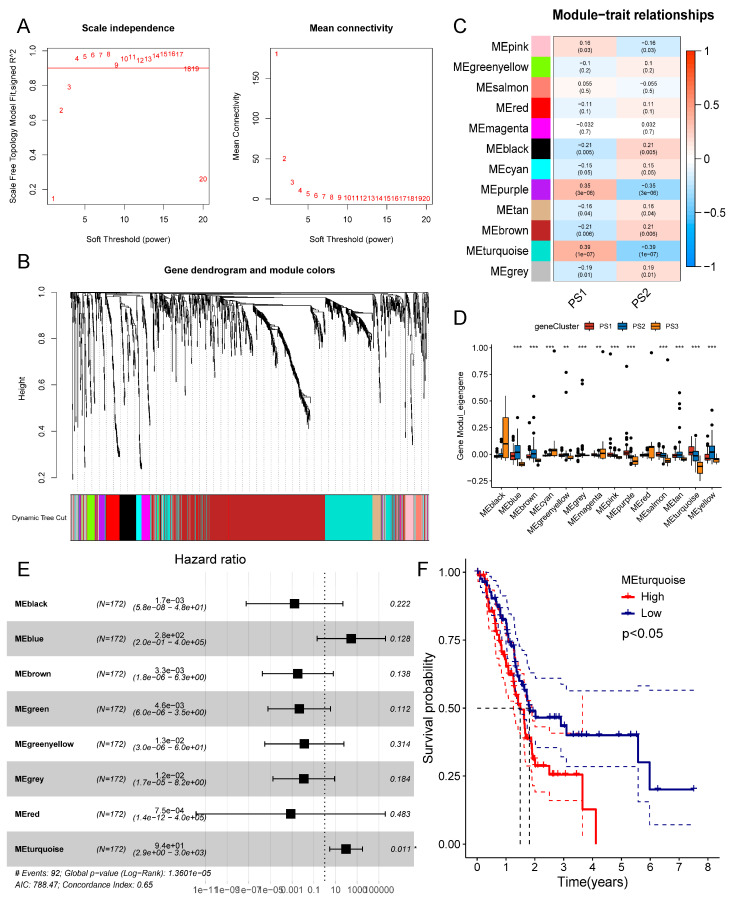
Identification of hub pyroptosis-related genes in PAAD. (**A**) Scale-free fit index and mean connectivity for various soft threshold powers to identify optimum soft threshold. (**B**) Gene dendrogram and module colors obtained from WGCNA. (**C**) Eleven modules gained from WGCNA. (**D**) Comparison of pyroptosis subtype scores in different modules. (**E**) Multivariate Cox analysis of scores in different modules. (**F**) Kaplan–Meier curves of OS classified by the turquoise module scores. * *p* < 0.05, ** *p* < 0.01, *** *p* < 0.001.

## Data Availability

Data are contained within the article and Appendix A.

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
