# Peer review of "Identification of Novel Tumor Pyroptosis-Related Antigens and Pyroptosis Subtypes for Developing mRNA Vaccines in Pancreatic Adenocarcinoma"

_biomedicines, 2024, doi:10.3390/biomedicines12040726_

Round 1

Reviewer 1 Report

Comments and Suggestions for Authors

In the manuscript “Identification of novel tumor pyroptosis-related antigens and pyroptosis subtypes for developing mRNA vaccines in pancreatic adenocarcinoma” authors integrate RNA sequencing data and compare genetic alterations, and analyze relationships between immune cell abundance and tumor antigens to identify pyroptosis-related antigens for mRNA vaccine development and discern eligible candidates for vaccination. Authors propose that the overexpressed and mutant pyroptosis-related genes associated with poor prognosis, should be targeted with mRNA vaccines.

Authors identify four overexpressed and mutant pyroptosis-related genes associated with poor prognosis.

The protein products of these genes are proposed to induce pyroptosis and to trigger immune responses.

However, the overexression of these genes is associated with poor prognosis.

Furthermore, although higher tumor mutation burden and somatic cell mutation rate correlated with stronger anti-tumor immunity, the subtypes with the higher mutation burden and somatic cell mutation rate, ps1 and ps2 had the worse overall survival.

In summary, PS2 and PS1 subtypes are proposed to represent immunologically “hot” tumor while PS3 subtype was an immunologically “cold” tumor. This again points to a critical target. However, why is the “cold tumor” correlated with the better chance to survive?

This could be explained by factors other than the pyroptosis. If not, then stronger emphasis needs to be placed to present the findings in a way that is easier to comprehend the logic. Authors need to be more rigorous in the choice of the material to present.

Excessive material should be placed to the supplement.

Another aspect that needs attention is the use of sentences that are incomplete. Example is “we established mRNA vaccines and searched for potentially benefitial PAAD patients in this article.”

It might be better to write “we established potential targets for mRNA vaccines and searched for PAAD patients that may benefit from those vaccines”.

In conclusion the manuscript needs a clearer focus and a rigorous presentation.

Comments on the Quality of English Language

None.

Reviewer 2 Report

Comments and Suggestions for Authors

This study presents an intriguing approach to the development of mRNA vaccines for pancreatic adenocarcinoma (PAAD). It systematically explores the selection of mRNA vaccine candidates based on pyroptosis-related antigens and identifies pyroptosis subtypes. The paper is overall clear, well-structured, and employs a variety of experiments and analyses effectively.

1. Clear emphasis on the clinical significance and novelty of the study is needed.

2. Connecting the results to the clinical prognosis based on pyroptosis subtypes would enhance the completeness of the paper.

3. Adding explanations for graphs or statistical results that require interpretation would improve reader understanding.

This paper is regarded as a promising contribution to future research in mRNA vaccine studies for PAAD, introducing novel indicators for consideration.

Comments on the Quality of English Language

The overall quality of English language in the manuscript is quite good. The sentences are well-structured, and technical terms are appropriately used. 

Reviewer 3 Report

Comments and Suggestions for Authors

The present work submitted to me for review is, on the one hand, very interesting. On the other hand, it contains several aspects that need to be explained before it can be accepted for publication.
1) Why do you only use databases? I am very sceptical about papers where the authors do not have their own clinical material? In vitro tests on cell lines would also be necessary here.
2) Please provide a schematic representation of all results, this will make it much easier to understand the issues you are dealing with.
3 I think the discussion section is very weak - there is no clinical reference, which is very important.

Round 2

Reviewer 3 Report

Comments and Suggestions for Authors

Unfortunately, the revised version of the text lacks a response to my comments and remarks. Please complete and highlight in colour.

Round 3

Reviewer 3 Report

Comments and Suggestions for Authors

After checking, I see that the authors have closed the issue regarding my comments. it is ok.